# CmPn/CmP Signaling Networks in the Maintenance of the Blood Vessel Barrier

**DOI:** 10.3390/jpm13050751

**Published:** 2023-04-28

**Authors:** Revathi Gnanasekaran, Justin Aickareth, Majd Hawwar, Nickolas Sanchez, Jacob Croft, Jun Zhang

**Affiliations:** Department of Molecular and Translational Medicine (MTM), Texas Tech University Health Science Center El Paso, El Paso, TX 79905, USA

**Keywords:** cerebral cavernous malformations, CCM signaling complex, blood-brain barrier, tight junctions and adherens junction, endothelial cells, progesterone, classic nuclear and non-classic membrane PRG receptors

## Abstract

Cerebral cavernous malformations (CCMs) arise when capillaries within the brain enlarge abnormally, causing the blood–brain barrier (BBB) to break down. The BBB serves as a sophisticated interface that controls molecular interactions between the bloodstream and the central nervous system. The neurovascular unit (NVU) is a complex structure made up of neurons, astrocytes, endothelial cells (ECs), pericytes, microglia, and basement membranes, which work together to maintain blood–brain barrier (BBB) permeability. Within the NVU, tight junctions (TJs) and adherens junctions (AJs) between endothelial cells play a critical role in regulating the permeability of the BBB. Disruptions to these junctions can compromise the BBB, potentially leading to a hemorrhagic stroke. Understanding the molecular signaling cascades that regulate BBB permeability through EC junctions is, therefore, essential. New research has demonstrated that steroids, including estrogens (ESTs), glucocorticoids (GCs), and metabolites/derivatives of progesterone (PRGs), have multifaceted effects on blood–brain barrier (BBB) permeability by regulating the expression of tight junctions (TJs) and adherens junctions (AJs). They also have anti-inflammatory effects on blood vessels. PRGs, in particular, have been found to play a significant role in maintaining BBB integrity. PRGs act through a combination of its classic and non-classic PRG receptors (nPR/mPR), which are part of a signaling network known as the CCM signaling complex (CSC). This network couples both nPR and mPR in the CmPn/CmP pathway in endothelial cells (ECs).

## 1. Introduction

Cerebral cavernous malformations (CCMs), among the most common brain vascular malformations, are characterized by abnormally dilated intracranial capillaries resulting in the destruction of the blood–brain barrier (BBB), leading to hemorrhagic stroke [1,2]. Three CCM genes, KRIT1 (CCM1) [3,4], MGC4607 (CCM2) [5,6], and PDCD10 (CCM3) [7], have been identified as causes of familial CCM (fCCM); at least one of these genes is disrupted in most human fCCM cases [1]. It has been defined that the three CCM proteins form the CCM signaling complex (CSC) that maintains the integrity of the blood–brain barrier (BBB) [8,9,10,11]. Although the majority of carriers of CCM gene mutations are asymptomatic, this autosomal dominant disorder with incomplete penetrance can lead to irreversible brain damage once symptoms manifest, typically in the form of focal hemorrhage. Despite ongoing research, the specific molecular mechanism that triggers the development of CCM pathology remains unclear.

The BBB, the blood vessel barrier in the brain, is a complex, dynamic, and selective interface between the bloodstream and central nervous system (CNS) [12,13,14]. The BBB is responsible for regulating the transfer of molecules between the bloodstream and the central nervous system (CNS) [15,16]. It acts as a barrier, blocking the entry of certain blood components, such as pathogens and drugs, into the cerebrospinal fluid (CSF) or CNS. However, it permits the selective passage of vital nutrients (glucose, amino acids, and lipid-soluble substances), hormones (neurosteroids), and signaling agents. It is formed by three structural features: tight junctions between non-fenestrated capillary endothelial cells, pericytes, basement membranes, and astrocyte endfeet (AE) [12,13,14]. The disruption of endothelial cell junctions within the blood–brain barrier (BBB) can lead to hemorrhagic stroke with high morbidity and mortality [17,18,19]. Likewise, any disruption to the four essential components of the BBB, namely, tight junctions between non-fenestrated capillary endothelial cells, pericytes, basement membranes, and astrocyte endfeet, can compromise the integrity of the BBB. Although there are structural and functional deviations between the peripheral blood vessels and the blood–brain barrier, and variations in their signaling pathways have been reported [13,20,21,22,23], there are also numerous molecular and cellular mechanisms shared between them, even though many questions remain unanswered [22,24,25]. Additionally, the two main lesion sites in CCM pathology have been identified in both the BBB and cutaneous peripheral blood vessels [1,26]. Therefore, in this review, we provide a brief description of the underlying mechanisms of vessel leakage between peripheral blood vessels and the BBB, with an emphasis on hemorrhages (equivalent to symptomatic CCMs for neurosurgery) related to the BBB, due to limited space.

It is crucial to comprehend the molecular mechanisms underlying the signaling pathways that regulate and maintain blood–brain barrier permeability in both physiological and pathological states. The elucidation of the molecular mechanisms involved in the disruption of the BBB can provide insights into the effects of various factors, including abusive drugs, toxins, and pathogens. This knowledge can also be instrumental in understanding numerous neurological pathologies, such as encephalitis, meningitis, and, in particular, hemorrhagic stroke [27,28]. The maintenance of blood–brain barrier (BBB) integrity primarily relies on two crucial endothelial cell junctions: adherens junctions (AJs) and tight junctions (TJs). These junctions are formed by different molecular components, but they are functionally and structurally interconnected [29]. This review exclusively examines the pathological leakage resulting from disturbed endothelial cell contacts in the disrupted BBB. While transport pathways across the BBB, such as receptor-mediated endocytosis, exosomal transport, and transcytosis of endothelial cells in the BBB [30,31,32,33], have immense potential for therapeutic use [34,35,36,37], they could not be covered in this review due to space constraints.

Steroids, which are either derived from dietary sources or synthesized within the body, are defined by their distinctive four-ring carbon structure and play vital roles as hormones in various physiological functions. These compounds have been employed in numerous therapeutic applications for a broad array of human disorders [38,39,40], such as vascular conditions, although their use continues to be a subject of debate [41]. Specifically, the importance of progesterone (PRG), a key female sex steroid, in preserving the integrity of the blood–brain barrier has been the focus of extensive research. The latest research suggests that the reciprocal modulation and coupling of both the classic nuclear PRG receptor (nPR) and the non-classic membrane PRG receptor (mPR) signaling occur through the CSC, which is a crucial modulator of the BBB [8,9,10,42,43,44,45,46]. According to these findings, a novel signaling network called the CSC-mPR-PRG-nPR/CSC-mPR-PRG (CmPn/CmP) operates within endothelial cells (ECs), while the CmPn signaling network is present in nPR(+) cells, and the CmP signaling network is also present in nPR(−) cells (Figure 1).This network is subject to dynamic modulation and fine-tuning through a range of feedback mechanisms under the influence of PRG actions, which are crucial for maintaining the BBB [8,46].

At present, tissue plasminogen activator (tPA) is the most effective medication recommended for treating ischemic strokes. tPA helps dissolve blood clots, restoring blood flow to the brain. Surgery remains the only viable option for treating hemorrhagic strokes, highlighting the importance of preventing and managing strokes [47,48,49,50]. While there is currently no available clinical agent or method for effectively repairing the blood–brain barrier (BBB), several epigenetic mechanisms and regulators that can either protect or disrupt the BBB have been identified. This suggests that the identification of both extracellular and intracellular factors in the future could hold therapeutic potential for addressing this challenge [51].

This article provides a comprehensive review of the interplay among steroids, EC junctions, and blood vessel permeability, including the impact of inflammation on BBB integrity and the effects of angiogenic factors on BBB integrity. It also discusses the potential of steroids as future therapeutics and examines the effects of sex steroids on neovascularization and downstream angiogenic factors of PRG signaling. Furthermore, it highlights PRG’s neuroprotective effects on the BBB and the actions of PRG and its derivatives as neurosteroids. Lastly, the article explores the impact of PRG-mediated signaling on endothelial cell function and the maintenance of BBB integrity by the newly defined CmPn/CmP signaling networks.

## 2. Key Factors Influencing Blood Vessel Permeability

Blood vessel permeability can be influenced by various factors, such as the integrity of endothelial cell junctions, the occurrence of inflammatory events, the expression of adhesion molecules, the activation of intracellular signaling pathways, and the release of angiogenic factors. Additionally, the presence of vascular pores or fenestrations, vesicular transport, and cell–cell communications can play a role in regulating blood vessel permeability. In simple terms, the permeability of blood vessels is controlled by molecular processes that regulate the EC junction to maintain stability in the vasculature [52,53]. Impaired EC stability contributes to various health issues. Similarly, numerous physiological and environmental factors, such as blood flow dynamics [54], heparin and glycocalyx [55,56], basement membrane [57], and pathological conditions including perturbed angiogenesis and lymphangiogenesis [58,59], cancers [58], diabetes [60], and aging [61,62], can all impact EC junction responses and result in increased permeability of blood vessels.

### 2.1. EC Junctions

As mentioned earlier, EC junctions mainly consist of TJs and AJs [63]. TJs consist of junction adhesion molecules (JAMs), claudins, and occludins [64], whereas AJs contain cadherin components such as VE-cadherin and N-cadherin [65]. Changes to the composition of both adherens junctions (AJs) and tight junctions (TJs) can potentially interfere with normal vascular permeability by affecting the transportation of molecules between adjacent cells. This can lead to a compromised vascular barrier, allowing immune cells and other substances to leak through. Endothelial cell junctions enable the flow of blood proteins, mediators, and immune cells through the vasculature. The regulation of endothelial cell permeability involves different mechanisms such as vascular pores or fenestrations, vesicular transport, and cell–cell communication [63]. Furthermore, damaged endothelial cells can lead to increased permeability of inflammatory cytokines and other factors such as vascular endothelial growth factor (VEGF), thrombin, and histamine [66,67]. If increased blood vessel permeability in the brain is present and reabsorption is not functioning adequately, this can result in the development of edema, which may worsen the conditions, leading to BBB dysfunction. Such complications can cause serious brain injuries, including hemorrhagic strokes [63].

### 2.2. Inflammation and BBB Integrity

Inflammation is widely recognized as a significant contributor to the BBB dysfunction [68,69]. Leukocytes can firmly attach and migrate through the ECs by utilizing the vascular cell adhesion molecule 1 (VCAM-1) and intercellular adhesion molecule 1 (ICAM-1) [70]. When there is EC damage or EC junction dysregulation, immune cells release proinflammatory cytokines, which activate the endothelium and allow for selectin attachment to leukocytes [71]. The attachment and migration of leukocytes through the endothelium, facilitated by VCAM-1 and ICAM-1, can result in a delay in leukocyte circulation. This delay provides an opportunity for chemokines on the endothelial surface to interact with the leukocytes and activate integrin [72].

### 2.3. Angiogenic Factors and BBB Integrity

VEGF plays a crucial role in regulating vascularization from embryogenesis to adulthood, including the formation of adult blood vessels [73,74]. In mammals, there are three VEGF tyrosine kinase receptors that can bind to five VEGF ligands [73]. When VEGF receptors associate with their ligand, and this complex interacts with VE-cadherin, a component of AJs, BBB permeability can increase [75]. In addition, VEGF can induce the expression of integrin and selectin to initiate an inflammatory response [76].

## 3. Angiogenic Impacts of Steroids on Maintenance of Vasculature

### 3.1. Steroids Can Influence the BBB Integrity

Steroids have been found to influence the BBB through their regulation of VCAM-1 expression [77]. ESTs have been found to reduce the expression of VCAM-1 mRNA and protein, which is induced by lipopolysaccharide (LPS) and inhibits monocyte adhesion to ECs. In contrast, glucocorticoids (GCs), a related steroid, have the opposite effect by stabilizing VCAM-1 mRNA expression [77]. These findings suggest that the administration of GCs to ECs in the central nervous system (CNS) could potentially enhance the preservation of blood–brain barrier (BBB) integrity in various CNS disorders, particularly in cases where the BBB has been compromised [29]. Glucocorticoids (GCs) mainly exert their actions by binding their GC receptors (GRs), and they occasionally also bind to the mineralocorticoid receptor (MR) [78]. It has been reported that GCs increase the expression levels of occludin, claudin-5, cadherin-9, and VE-cadherin, key components of EC intercellular TJs and AJs, at both transcriptional and translational levels [29,41,79,80,81,82,83]. For example, GCs are known to induce the expression of occludin in the brain endothelium at the transcriptional level by binding to GC-responsive elements (GREs) in the occludin promoter [29,41,80,81,82], indicating their potential in restoring BBB integrity. However, the well-known fact that VCAM-1 promotes leukocyte attachment [84,85,86], has raised doubts regarding the effectiveness of GCs in maintaining endothelial cell junctions.

GCs are able to specifically have effects on the vasculature upon binding GRs through nitric oxide biosynthesis [87,88], further transmitting a signaling pathway that causes inflammation in ECs and affects angiogenesis [87,88]. The vascular inflammation caused by GCs has recently been found to be related to hypertension in mouse model studies [89]. Additionally, studies have also found that hypertension is in part mediated by EC GR expression [90]. When an individual undergoes a stress response, GC is released. As a ubiquitously expressed steroid receptor, GRs have two alternative splicing isoforms, dominant GRα and the less common GRβ, which drive a large spectrum of responses across various cell types and tissues [29,83,91,92]. GRβ has intrinsic activities and regulates a collection of genes related to inflammatory processes, cell communication, migration, and proliferation [29,83,92,93]. GRβ was also found to play a major role in maintaining BBB integrity [83]. Since evidence has suggested that GC/GR actions can improve BBB integrity by increasing the expression levels of many key components of TJs and AJs [29,41,79,80,81,82,83], attempts have been made to explore the therapeutic potential of GC/GR actions on neurovascular ECs in improving clinical symptoms of hemorrhagic strokes. However, a large clinical trial on the therapeutic efficacy of GC/GR actions on hemorrhagic strokes provided contradicting outcomes [94,95], as described below.

### 3.2. Steroids Might Play Major Roles for Compromised BBB

ESTs belong to another group of important regulatory hormones of BBB permeability. They protect the BBB before menopause, but may increase BBB permeability with aging [96,97,98]. Following the initial discoveries regarding the positive effects of EST treatment in stroke prevention, subsequent clinical evidence presented conflicting results regarding the impact of ESTs on blood–brain barrier (BBB) integrity [99,100], These findings highlighted the significant dependence of EST efficacy on the reproductive or postmenopausal age of women [99,100]. For example, hormone (mixture of ESTs + PRGs) replacement therapy (HRT) appears to increase stroke risk and worsen neurological outcomes [99,101,102,103,104]. More specifically, hormonal changes during pregnancy appear to be a major risk factor for stroke in women [105,106,107,108,109,110,111,112], suggesting that the EST-to-PRG ratio may be critical for hemorrhagic stroke susceptibility. Additionally, hemorrhagic stroke is the most dominant type (up to 74%) of strokes during pregnancy [113,114,115,116,117,118,119,120]. Although pregnancy as a hemorrhage risk factor in women with CCMs is under debate [121], CCM patients are more likely to bleed or to have lesion expansion during pregnancy, suggesting a potential influence of sex hormones in hemorrhagic events [8,46,122]. Therefore, clinical recommendations for asymptomatic CCM patients with pregnancy were made with conservative monitoring by magnetic resonance imaging (MRI) [123].

Evaluation of female hormone contraceptives, such as PRG and its derivatives, are yet to be explored for effects on hemorrhagic stroke management and prevention. A recent report indicated that mifepristone (MIF), a widely used contraceptive and a known antagonist of PRG-nPR signaling, can inhibit PRG actions through nuclear PRG receptors (nPRs) in cells that express them. However, in cells that lack nPR expression, MIF can function as a PRG agonist on its own or in combination with PRG to activate PRG signaling through non-classic membrane PRG receptors (mPRs), leading to specific mPR-mediated PRG actions [8,42,43,44,46,122,124,125].

Hence, MIF and PRG collaborate to amplify mPR-specific PRG action in this signaling cascade (Figure 1). Within nPR(−) ECs, the CmP signaling network reduces CSC stability through the adverse effects of mPR-specific PRG signaling actions on the CSC [PRG, MIF alone or combined (PRG/MIF)], in the absence of the positive impacts of nPR-specific PRG signaling actions (Figure 1) [8,42,43,44,45]. The balance between classic and non-classic PRG signaling impacts the CSC function and recognizes the CSC as an important mediator of nPR and mPR crosstalk in nPR(+) cells [8,42,43,44,45]. This observation is supported by a previous finding that activation of mPR-specific PRG signaling can potentiate expression of the hormone-activated nPR-2 isoform [126]. This observation was later duplicated in nPR(−) microvascular ECs, indicating existence of a common CmP signaling network in various nPR(−) cells [45]. Several studies have suggested that the ratio of estrogens (ESTs) to progestogens (PRGs) could have a more substantial impact on blood–brain barrier (BBB) maintenance [105,106,107,108,109,110,111,112,113,114,115,116,117,118,119,120]. Additionally, clinical investigations have provided further proof of this by demonstrating a link between prolonged exposure to PRGs, as seen in birth control regimes, and the development of more severe hemorrhagic cerebral cavernous malformations (CCMs) [46,127].

### 3.3. Steroids as Potential Therapeutics

When damage to the endothelium takes place, EST aids in repairing the EC by reestablishing the tight and adherens junctions, which results in a decreased likelihood of hemorrhagic stroke. This finding has contributed to the utilization of tamoxifen in treating vascular conditions [128,129]. A similar finding of GC/GR actions to improve BBB integrity has been employed as a medical therapy to treat vascular malformations [29,41,79,80,81,82,83]. This approach was first applied in the treatment of infantile hemangiomas, but with varying success rates [130,131]. Similar endeavors have also been made on cerebral cavernous malformations (CCMs). A study on animals revealed that GC therapy can effectively tackle the bleeding characteristics in Ccm mice by targeting GRs in the brain’s endothelial cells [132]. Subsequently, various crucial molecular and cellular processes were analyzed, and researchers also investigated the potential clinical uses of therapies derived from the effects of GC/GR interactions on CCMs [133,134]. However, despite numerous efforts, clinical trials have not shown evidence to support the use of GC/GR-based therapies in treating CCMs [133,134,135,136,137,138].

Like ESTs and GC, the effect of PRG and its metabolites/derivatives (defined as PRGs), as well as contraceptive derivatives (progestins), on BBB integrity has also been primitively explored, despite a lack of understanding that different progestins used in contraception exhibit differential off-target effects via steroid receptors [139]. PRGs were indeed subsequently found to be able to bind both GRs and MRs [140,141]. It was reported that PRGs can repress the inflammatory response through their binding to GRs in a dose-dependent manner at concentrations within the physiologically relevant range [141]. This suggests the same positive effect of PRGs on the BBB. Furthermore, the neuroprotective properties of PRGs against VEGF, a major angiogenic factor that can lead to vascular leakage, have been thoroughly investigated [142,143,144]. Many contradicting results were presented; it was found that, unlike EST, which mainly promotes neovascularization through activating classic ER-mediated signaling pathways, PRGs regulate a variety of downstream factors which can be either angiogenic or antiangiogenic [145]. This suggests that PRGs might exert their cellular effect on ECs through different receptors. These findings suggest that PRGs may exert their cellular effects on ECs through various receptors. While no correlation was identified between the parameters measured in classic nuclear progesterone receptors (nPRs) and neovascularization, there is strong evidence to suggest that progestin-responsive genes may solely exert their effects through a non-classical mechanism [142,143,144,146]. In fact, one report demonstrated that PRG stimulation of NO, through increased eNOS expression in human vascular ECs does not involve nPRs, but instead employs non-classical mPRα through the PI3K/Akt and MAPK pathways [147].

## 4. Impacts of Sex Steroids on Maintenance of Vasculature

### 4.1. Neovascularization

Neovascularization is the formation of new blood vessels, which typically occurs in specific physiological processes, such as the menstrual cycle and tumorigenesis [148,149]. Disruption of angiogenic regulation can lead to the acquisition of enhanced angiogenic capabilities by endothelial cells (ECs) in the vascular beds, which can result in the formation of new blood vessels from destabilized sites in the basement membrane [149]. Sex steroids, particularly PRGs, regulate neovascularization in a diverse fashion. Several studies have found a phenomenon that low concentrations of PRG potentially promote neovascularization through nPRs in a dose-dependent fashion, while high concentrations inhibit the original effect [150,151,152].

### 4.2. Downstream Angiogenic Factors of PRG Signaling

The uterus vasculature is sensitive to cytokines, growth factors, and sex hormones, affecting angiogenesis and tissue growth. VEGF distribution changes during the menstrual cycle may impact angiogenesis. Endometrial stromal cells and myometrium smooth muscle cells produce VEGF, which increases in rat uterine tissue due to PRGs and ESTs [153,154,155]. Glandular VEGF immunostaining increases during the menstrual cycle, while nPRs decrease and ERs remain unchanged. Contraceptives containing PRG and EST lead to decreased VEGF without affecting menstrual bleeding or EC density, supporting PRG regulation of glandular endometrial VEGF levels. Other regulators of neovascularization include bFGF, PD-ECGF, Ang, HIF1a, and NOS.

Moreover, research has demonstrated that PRGs can increase the actions of nitric oxide (NO) in vascular ECs exclusively through membrane progesterone receptor (mPR) signaling. In a study of human umbilical vein endothelial cells (HUVECs), the role of mPRs in angiogenesis was investigated [156]. As an mPR agonist, PRG–BSA conjugates increased NO levels in HUVECs, indicating that progestin-responsive genes are activated by mPRs at the cell membrane. The study also revealed that PRG–mPR interactions boost eNOS activity via eNOS phosphorylation [157]. Phosphoinositide 3-kinase (PI3K)/Akt and mitogen-activated protein kinase (MAPK) inhibitors reversed this stimulation of eNOS phosphorylation brought on by the PRG–mPR action. Similarly, siRNA silencing of mPRα diminished the stimulatory signaling effects. However, knockout of nPR did not result in a similar blockage of stimulatory effects [158]. Therefore, mPRα-mediated signaling via PI3K/Akt and MAPK signaling facilitates the PRG-induced stimulation of NO synthesis in HUVECs. The abnormal expression of eNOS [159] and MMPs [160,161] in CCM-deficient mutant models has been widely investigated using systems biology methods [10] in both in vitro and in vivo models.

## 5. Angiogenic Impacts of PRG on the Neurovascular Unit (NVU)

### 5.1. Neuroprotective Impact of PRGs on the NVU

The NVU is composed of neurons, astrocytes, ECs, pericytes, microglia, and basement membranes [13,23,52,57]. PRG is known to have neuroprotective effects on neurons vulnerable to ischemic and excitotoxic damage [162,163,164]. Post-traumatic brain injury (TBI) edema can be a consequence that leads to excitotoxic effects on neurons. The administration of PRG has been found to decrease neuronal death and edema, resulting in improved behavioral recovery, as reported in studies [165,166]. Furthermore, reduced antioxidant enzyme activity leads to oxidative stress in neural tissues. A study demonstrated that low doses of PRG and/or EST can significantly increase superoxide dismutase activity and mitigate the negative impact of lipid peroxidation [167]. Lipid peroxidation, which can initiate cerebral edema, is a common occurrence after brain trauma. A study demonstrated that rats treated with PRG exhibited one-third of the lipid peroxidation observed in the controls [168].

In addition, PRG is known to regulate microglial activation, reduce the production of proinflammatory cytokines, increase the expression of antiapoptotic Bcl-2 protein, promote myelinization and myelin repair [169], and regulate neuronal plasticity via neurotrophins such as brain-derived neurotrophic factor (BDNF) by interacting with one class of non-classic PRG membrane receptors, PRG receptor membrane component 1 (PGRMC1) [170,171,172]. Therefore, neurological effects of long-term exposure to contraceptives (mixture of ESTs and PRG derivatives) on cognitive function have recently been noticed [173], although whether ESTs and PRGs coordinate or counteract during these neuroprotective steroid actions remains under debate [172].

### 5.2. Neuroprotective Effects of PRG and Its Derivatives on the BBB against Thrombin and Matrix Metalloproteinase (MMP)

Thrombin has the ability to impair both endothelial cell (EC) junctions and the extracellular matrix (ECM) [174,175]. Thrombin can compromise the BBB’s integrity in two ways. Firstly, it activates proteases via protease-activated receptor 1 (PAR-1), resulting in the internalization of the protein Claudin-5 present in the BBB’s tight junctions (TJs) [175]. Additionally, thrombin can cause necrosis and apoptosis in the brain due to ischemia and/or hypoxia, which can result in cerebral edema, stroke, and other CNS injuries [175,176]. 

Activated MMPs have been linked to a compromised blood–brain barrier (BBB). This is believed to be due to their ability to affect the integrity of the BBB by breaking down and restructuring the extracellular matrix that surrounds the capillaries in the brain, as well as by degrading the proteins responsible for tight junctions [177,178,179]. PRG has demonstrated neuroprotective properties by reducing the induction of MMP and edema in the BBB [180]. PRG and its derivatives [allopregnanolone and 5α-dihydroprogestrone (5α-DHP)] regulate MMPs through mPRs as neurosteroids. The inhibition of PRG conversion to 5α-DHP and allopregnanolone has been shown to block PRG’s neuroprotective effects [175]. Furthermore, PRG has been shown to partially inhibit the actions of thrombin in the myometrium [176,181].

### 5.3. Neuroprotective Effects of PRG on BBB against Inflammatory Pathway

Like EST, PRG can attenuate LPS-stimulated inflammation in a dose-dependent fashion by suppressing the LPS-induced NF-κB activation. These specific anti-inflammatory effects of PRG can be inhibited by MIF, suggesting PRG-nPR as a sole anti-inflammatory signaling pathway [182].

PRG can potentially regulate intracellular free calcium (Ca^2+^) levels by activating a specific membrane receptor called PGRMC1. This activation results in reduced nuclear accumulation of Ca^2+^-dependent nuclear factor of activated CD8^+^ T cells 1 (NFAT1), associated with T-cell development and activation. Therefore, PRG and its non-classic membrane receptor, PGRMC1, may directly suppress T-cell activation in inflammatory responses, indicating their involvement in non-classic PRG signaling [182]. PRG can also indirectly inhibit T-cell activation by stimulating IL-10-producing dendritic cells, while also inhibiting Th1 and Th17 differentiation of CD4^+^ T helper (Th) cells, shortening the active course of some chronic autoimmune inflammatory diseases in the brain [182].

### 5.4. Actions of PRG and Its Derivatives as Neurosteroids

Physiological impacts of PRG can be influenced by both mPRs and nPRs [183]. nPR-A and nPR-B are the most common nPR isoforms that mediate the response of PRG [163]. These nPRs are structurally different and have tissue-specific responses due to factors such as post-translational changes, in addition to various cofactors that lead to distinct responses for each isoform receptor [184]. Furthermore, mRNA transcripts of both nPR-A and nPR-B isoforms are found to colocalize in areas of the brain where nPRs are already known to be located [184]. Within the CNS, steroid levels are due to both secretions of endocrine glands and local metabolism. All neurons and glial cells can metabolize PRG; therefore, changes in the steroid hormone pool could indicate changes in the brain [185]. In fact, AP (3,5-tetrahydroxyprogesterone) is a PRG-derived neuro-metabolite that acts as an agonist for the receptor of neurotransmitter gamma-aminobutyric acid (GABA), the GABA-A receptor [186]. Therefore, PRGs have important physiologically antiepileptic, sedative, and regulatory effects on behavior and stress [186]. In sum, PRG and its derivatives, as neurosteroids, can indirectly influence the BBB integrity through their impact on neurons within the NVU.

## 6. Impacts of CmPn/CmP Networks on the BBB

### 6.1. The Impact of PRG-Mediated Signaling on Endothelial Cell (EC) Function in the Vasculature

PRG derivatives in hormone replacement therapy for postmenopausal women have complex cardiovascular effects due to various signaling pathways [187]. They promote endometrial safety and reduce blood pressure in hypertensive patients by increasing nitric oxide production via eNOS modulation; the cumulative impact of these PRG derivatives on the vasculature can ultimately provide protection against cerebral hemorrhage [188,189,190,191].

### 6.2. CmPn/CmP Signaling Ntworks Maintain the BBB Integrity

The intricate feedback regulation among the PRG-activated CmPn signaling network in nPR(+) cells can be summarized with a model of CSC-modulated classic and non-classic PRG signaling (Figure 1). As a sex steroid hormone, PRG binds to its nPR for classic PRG actions and mPR for non-classic PRG actions [125,192,193,194,195,196]. Additionally, recent findings demonstrated that CCM2 interacts with both CCM1 and CCM3 to form the CSC [1,10]. In this model, steroid actions are attained through the balanced efforts between nPR and mPR signaling pathways and are further fine-tuned by the CSC (Figure 1) [8,42,43,44,45,46,124,125]. The CSC can couple both classic nPR and non-classic mPR signaling to form the CmPn signaling network in nPR(+) cells and the CmP signaling network in nPR(−) cells [42,43,44,125,197]. The CmP signaling network was found to play a major role in maintaining BBB integrity in nPR(−) microvascular ECs within the NVU [8,46,125,198]. Additionally, within nPR(−) vascular ECs, the CSC is a master regulator governing homeostasis of PRG and its mediated signaling cascades in the more fragile CmP signaling network, indicating an important role of the CmPn/CmP signaling networks in angiogenesis and tumorigenesis [8,42,43,44,45,46,125,198]. Previous multi-omics data, which provided a global view of signal transduction modulated by the CSC, indicated that perturbed CSC leads to disruption of blood vessel cell junctions [8,46]. Comparative omics data across multiple models provided further supportive evidence that the perturbed CmP network in nPR(−) microvascular ECs leads to a compromised BBB with disrupted EC junctions, both in vivo and in vitro [8,45,46,198].

## 7. Conclusions

In this review, we attempted to summarize steroid signaling pathways that regulate BBB integrity through the impacts of certain steroid receptors on its key components, with special emphasis on the combined effect of PRG and CSC through the CmPn/CmP signaling network (Figure 1, Table 1).

The BBB is an intricate interface between blood flow and brain tissue with significant contribution to brain injuries, such as hemorrhagic stroke, through its disruption. Understanding the mechanisms through which various pathologies alter BBB integrity will allow for future advances in therapeutics for neurovascular injuries. Steroids, especially sex steroids (mainly ESTs and PRGs), have shown some major impacts on EC repair, neovascularization, and maintenance of BBB integrity. PRG is perhaps the most significant factor involved in maintaining vasculature and protecting neurons and the BBB. The CSC’s essential role in coupling and modulating nPR and mPR signaling was found to be integral in maintaining the BBB integrity within the CmPn/CmP signaling networks. Perturbation of this network, especially in CmP signaling in cerebral ECs, disrupts EC junction components and leads to compromised BBB. In addition to PRG, we described the role of other steroids in BBB maintenance, as well as potential roles in hemorrhagic stroke treatment.

## Figures and Tables

**Figure 1 jpm-13-00751-f001:**
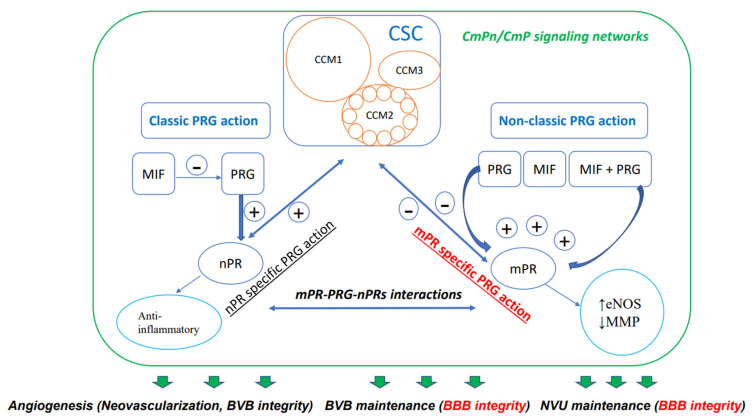
Diagram demonstrating CmPn/CmP signaling network (largest square) where the CSC (CCM signaling complex, CSC square) couples both classic nPR (classic PRG action) and non-classic mPR (non-classic PRG action) signaling to exert their influence on the performance of vascular ECs. Multiple small circles within CCM2 indicates its essential role with binding status with both CCM1 and CCM3, through its multiple alternative spliced isoforms. Dual-directional arrows indicate bi-directional effects; bold green arrows indicate eventual effects of the CmPn/CmP signaling network on vascular angiogenesis, including neovascularization and dynamic regulation of the blood vessel barrier. Within the green frame of the CmPn/CmP signaling network, dark-blue arrows and frames are major components of the network, while light-blue circles are downstream cellular events modulated by the network. Plus symbol (+) indicates positive impact while minus symbol (−) for negative effect. Within the CSC, light-brown circles represent key components: CCM1, CCM2, and CCM3. The signaling pathway colored in red is important for BBB integrity. As mentioned in the article, this signaling network also affects the preservation of the blood–brain barrier (BBB) and neurovascular unit (NVU), both of which are essential for maintaining BBB integrity.

**Table 1 jpm-13-00751-t001:** This table summarizes the steroid signaling pathways that regulate BBB integrity via their effects on key components of EC junctions, as well as relevant genes and key findings discussed in the text. Additionally, a list of recent publications discussing each topic in the text, as well as the newly discovered roles of CmPn and CmP signaling networks in maintaining the BBB among vascular ECs, is included.

Main Point	Key Findings	Genes Involved	Key References
Key factors influencing on the blood vessel permeability	The blood brain barrier (BBB, the blood vessel barrier) in the brain, is a dynamic and complex interface that tightly controls the molecular exchanges between the blood and central nervous system (CNS) the integrity of both BVB and BBB are determined by EC junctions that are made up of mainly adherens junctions (AJs) and tight junctions (TJs), which can be influenced by steroids.	JAMs, claudins, occludins, VE-cadherin, N-cadherin, VCAM-1, ICAM-1,	[15,16,27,28,73,75]
Angiogenic impacts of steroids on maintenance of vasculature	Steroids might play major roles on maintainence of BBB integrity, through their regulation of EC junction components, dysregulation of steroid-mediasted signaling pathway can lead to compromised BBB, therefore, steroids can be potential therapeutic targets.	MRs, GRs, GRα, GRβ, ERs, nPRs, mPRs, eNOS	[78,81,83,99]
Impacts of sex steroids on maintenance of vasculature	Sex steroids might play major roles on neovascularization, angiogenic signaling, and maintainence of BBB integrity, therefore, steroids can be potential therapeutic targets.	bFGF, PD-ECGF, HIF1a, Ang, NOS, eNOS, PI3Ks, Akt, MAPKs	[150,151,152,155]
Angiogenic impacts of progesterone (PRG) on the Neurovascular Unit (NVU)	PRGs have neuroprotective impact on the Neurovascular Unit (NVU)	BDNF, PGRMC1, Bcl-2, PAR-1, Claudin-5, MMP, NF-κB, IL-10, GABA-A receptor	[147,150,151,153]
Impacts of CmPn/CmP signaling networks on the BBB	PRG and its derivatives-mediated signaling pathways have major impacts on EC performance in vasculature. PRG and its derivatives-mediated CmPn/CmP signaling networks maintain the BBB integrity	CCM (1–3) genes, mPRs, nPRs,	[8,46,124,125,198]

## Data Availability

Not applicable.

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
