# Peer review of "CmPn/CmP Signaling Networks in the Maintenance of the Blood Vessel Barrier"

_jpm, 2023, doi:10.3390/jpm13050751_

Round 1
Reviewer 1 Report
The authors summarized steroid signaling pathways that regulate BBB integrity through the impacts of certain steroid receptors on its key components, with special emphasis on the combined effect of PRG and CSC through the CmPn/CmP signaling network, as shown in Fig. 1 and Table 1. It is a very unique and interesting review paper. The paper emphasizing the significance of CmPn/CmP signaling networks in the maintenance of the blood vessel barrier, should be accepted for publication in Journal of Personalized Medicine.
Author Response
Please see attached, thank you!

Reviewer 2 Report
The review manuscript by Gnanasekaran et al. attempts to address the widely-discussed and, at least in part, controversial topic of the influence of steroids on blood-brain barrier integrity and their possible significance in the treatment of cerebral cavernous malformations. The topic of this review is timely and could be of interest to readers. Unfortunately, this manuscript is unfocused and not particularly well-written, with numerous typos, grammar and/or wording issues. The article lacks flow and overall readability. Significant reorganization of the structure and writing style are required.
Major Comments:
- - The authors begin the introduction by emphasizing the importance of a better understanding of mechanisms for a damaged BBB in the treatment of CCM. However, the terms 'blood-brain barrier' and 'blood-vessel barrier' are used interchangeably throughout the article. Furthermore, the authors seek to draw comparisons between peripheral blood vessels and the BBB in terms of hemorrhage/ CCM therapy. Such a comparison, however, should be avoided because the vessels that form the BBB are considerably different from those found peripherally. (https://doi.org/10.1101/cshperspect.a020412, https://doi.org/10.1016/j.nbd.2009.07.030, https://doi.org/10.1186/s12987-019-0123-z).
- - The authors correlate a disrupted BBB only to damage/ hemorrhages. There is no mention of partial disruption of the BBB for therapeutic use. This field has vastly grown in recent years and is widely used in many current clinical trials.
(https://doi.org/10.1016/j.neo.2021.04.005, https://doi.org/10.2174/1381612822666151221150733)
- - The introductory part is not very detailed and briefly discusses the different types of BBB cells and the potential for steroidal treatments for BBB disruption. The introduction makes paracellular transport out to be the sole mode of solute transport into brain. Other significant transport processes, such as receptor-mediated endocytosis, exosomal transport, transcytosis,etc. are not mentioned.
- - Throughout the manuscript, the authors make claims and statements without appropriate citations. A few examples are lines line 195-197, 204-206, 222-223, 230-231, lines 262-263, 293, 306, 312, 332, 350 etc
- - Inconsistent writing. Example line46: mentions three essential components of the BBB, but mentions four.
- - Due to the lack of focus on a singular topic, the authors skip essential details in many sections. For example: Line 107: Key factors influencing blood vessel permeability. The authors fail to mention how diseased states, aging, concurrent medication and other environmental factors can also have a major role in modulating BBB permeability.
- - Line 94: While many conditions, such as stroke, can cause BBB disruption, the BBB can repair itself in 48-72 hours PMC5886838. The authors make it appear as if the BBB can never be repaired.
- - The authors correlate immune-cell requirement negatively for a disrupted BBB. However, in several diseases (tumors) and conditions, a partial immune response is therapeutically desired.
- - If the emphasis of the article is role of steroid mediated signaling in BBB repair; Lines 269-286 and 374-378 do not add to the article. .
The article's writing and grammar require significant revision. Please go over the content carefully; there are several grammar/wording issues that need to be addressed. Some instances are shown below.
- Inconsistent abbreviation: Line 36: abbreviation BBB has not been introduced. Line 53: BBB abbreviation is reintroduced.
- A number of non-descriptive assertions. Example: line 126-128, line 225: implies clinical use when it is not the case, line 233-235: variable outcome?
- - Line 8: the word destruction can be replaced by disruption
- - Line 11: missed pericytes and microglia (part of the NVU)
- - Quality of the figure needs to be improved
- - Inconsistent font used throughout the text example: line 16, lines 199-201
- - Grammar: example Lines 60-65
Author Response
Please see attached, thank you!

Round 2
Reviewer 2 Report
Appropriate changes made